# 3,4-*Seco*-Isopimarane Diterpenes from the Twigs and Leaves of *Isodon Flavidus*

**DOI:** 10.3390/molecules27103098

**Published:** 2022-05-12

**Authors:** Wan-Fei Li, Zheng-Ming Liang, Chen-Liang Zhao, Nga Yi Tsang, Ji-Xin Li, Ya-Hua Liu, Kang He, Lu-Tai Pan, Lijun Rong, Juan Zou, Hong-Jie Zhang

**Affiliations:** 1College of Pharmacy, Guizhou University of Traditional Chinese Medicine, Dongqing South Road, Guiyang 550025, China; li_wanfei@163.com (W.-F.L.); 18482082@life.hkbu.edu.hk (C.-L.Z.); lijixinmylove@yeah.net (J.-X.L.); lyh_1123@126.com (Y.-H.L.); hekang0851@163.com (K.H.); ltpan@sina.cn (L.-T.P.); 2School of Chinese Medicine, Hong Kong Baptist University, Hong Kong, China; 18482473@life.hkbu.edu.hk (Z.-M.L.); 12012173@life.hkbu.edu.hk (N.Y.T.); 3Department of Microbiology and Immunology, College of Medicine, University of Illinois at Chicago, 909 South Wolcott Avenue, Chicago, IL 60612, USA; lijun@uic.edu

**Keywords:** *Isodon flavidus*, diterpenoids, cytotoxicity, antifungal activity, antiviral activity

## Abstract

Three isopimarane diterpenes [fladins B (**1**), C (**2**), and D (**3**)] were isolated from the twigs and leaves of Chinese folk medicine, *Isodon flavidus*. The chemical structures were determined by the analysis of the comprehensive spectroscopic data, and the absolute configuration was confirmed by X-ray crystallographic analysis. The structures of **1**–**3** were formed from isopimaranes through the rearrangement of ring A by the bond break at C-3 and C-4 to form a new *δ*-lactone ring system between C-3 and C-9. This structure type represents the first discovery of a natural isopimarane diterpene with an unusual lactone moiety at C-9 and C-10. In the crystal of **1**, molecules are linked to each other by intermolecular O-H···O bonds, forming chains along the *b* axis. Compounds **1**–**3** were evaluated for their bioactivities against different diseases. None of these compounds displayed cytotoxic activities against HCT116 and A549 cancer cell lines, antifungal activities against *Trichophyton rubrum* and *T. mentagrophytes*, or antiviral activities against HIV entry at 20 µg/mL (62.9–66.7) µM. Compounds **1** and **3** did not show antiviral activities against Ebola entry at 20 µg/mL either; only **2** was found to show an 81% inhibitory effect against Ebola entry activity at 20 µg/mL (66.7 µM). The bioactivity evidence suggested that this type of compound could be a valuable antiviral lead for further structure modification to improve the antiviral potential.

## 1. Introduction

Natural products, especially plant-derived natural products, have long been an important source of molecules for drug discovery [1]. Diterpenes are one class of plant compounds that have been constantly investigated for their valuable potential in drug discovery due to their diversified structures and pronounced biological activities [2,3]. Isopimaranes are a subtype of diterpenes featuring carbotricyclic rings. Their rich stereochemistry features and broad biological activities render them the interesting molecules applicable to the pharmaceutical industry. They have been reported to own remarkable biological activities, including cytotoxic, antimicrobial, and anti-inflammatory activities [4,5,6,7,8]. The 3,4-*Seco*-isopimarane diterpenes belong to isopimaranes with the rearrangement of ring A by the bond break between C-3 and C-4. To date, there have been few reports on natural 3,4-*seco*-isopimaranes [9,10], and no antiviral activity has been reported for this type of compound.

The genus *Isodon* (formally *Rabdosia*) contains approximately 150 species of wild plants, and about 30 species have a long tradition of use as Chinese popular folk medicines [11,12]. *Isodon* species are rich in producing diterpenes with a diversity of carbon skeletons and a variety of biological activities [11,13,14]. Diterpenoids with a tricyclic core system and their *seco*-derivatives were also reported from this genus, including *I. rubescens*, *I. lophanthoides*, and *I. flavidus* [15,16,17,18]. *I.*
*flavidus* is a commonly used herb to defend against tinea pedis by the Miao minority in China [10]. Phytochemical investigation of this plant revealed the presence of isopirmarane [19,20] and *ent*-kaurane diterpenoids [15], along with other types of compounds such as flavonoids and steroids [19,20]. However, no biological evaluation of this medicinal plant was reported in these articles. In our previous study, two antifungal constituents, fladin A and lophanic acid, were isolated and identified from this plant. Fladin A was a 3,4-*seco*-isopimarane diterpenoid, and lophanic acid was an abietane diterpenoid [10]. In a search for novel bioactive compounds from *I.*
*flavidus*, we further obtained three new diterpenes, namely fladins B (**1**), C (**2**), and D (**3**). They also belong to 3,4-*seco*-isopimarane diterpenes but with a *δ*-lactone group formed between C-3 and C-9 (Figure 1). This article reports the isolation, structural identification, as well as biological evaluation (cytotoxicity against cancer cells, antifungal activity against *Trichophyton* fungi, and anti-Ebola entry activity) of these novel 3,4-*seco*-isopimarane diterpenes from the plant *I.*
*flavidus*.

## 2. Results

Plants from the genus *Isodon* are claimed to be a rich source of bioactive diterpenes with a diversity of carbon skeletons. The folk medicinal uses of *I. flavidus* [21] and our previous discovery of antifungal diterpenes from *I. flavidus* [10] gave rise to our great interest in this medicinal plant. Our ongoing research of the methanolic extract of the twigs and leaves of *I. flavidus* led to the identification of three new 3,4-*seco*-isopimarane diterpenes.

### 2.1. Chemistry and Structure Elucidation

Compound **1**, [α${]}_{D}^{28}$ −8.59° (*c* 0.33, CHCl_3_), was obtained as colorless cubic crystals, and the molecular formula was determined to be C_20_H_30_O_3_ by positive HR-ESI-MS ([M + H]^+^ *m*/*z* 319.2269, calcd. 319.2273). The IR spectrum of **1** displayed a hydroxyl group (*ν*_max_ 450 cm^−1^) and a *δ*-lactone group (*ν*_max_ 1694 cm^−1^). Evidenced from the ^1^H and ^13^C NMR (Table 1), DEPT spectral, and the HSQC correlation data, the structure of **1** showed signals of four tertiary methyls [*δ*_H_ 0.88, 1.14, 1.28, and 1.33 (each 3H, s); *δ*_C_ 19.9, 21.2, 26.4, and 34.4], a vinyl [*δ*_H_ 5.80 (1H, dd, *J* = 17.5, 10.7 Hz), 4.89 (1H, dd, *J* = 10.7, 1.2 Hz), 4.94 (1H, dd, *J* = 17.5, 1.2 Hz); *δ*_C_ 149.2 and 110.2], one trisubstituted carbon–carbon double bond [*δ*_H_ 5.57 (1H, dt, *J* = 5.6, 2.1); *δ*_C_ 127.5 and 132.8], six methylenes, one methine [*δ*_H_ 2.07 (1H, m); *δ*_C_ 43.8], two oxygenated quaternary carbons (*δ*_C_ 74.6 and 87.0), two sp^3^ aliphatic carbons (*δ*_C_ 37.8 and 21.2), and one carbonyl carbon (*δ*_C_ 172.7). These data indicated **1** to be a diterpenoid.

According to the HMBC spectral data (Figure 2) of **1**, the presence of correlations from H-15 to C-12, -13, -14, and -17, from H-16 to C-13 and -15, and from H-17 to C-12, -13, -14, and -15 indicated the vinyl group at C-13. The presence of correlations from H-7 to C-6 and -8 assigned the trisubstituted carbon–carbon double bond formed between C-7 and C-8. The correlations observed from H-5 to C-4, -6, -10, -18, -19, and -20, and from H-20 to C-1, -5, -9, and -10 assigned the two oxyquaternary carbons at C-4 and -9, respectively. The correlations from H-2 to C-1 and -3, from H-1 to C-3, -9, and -10, and from the methyl signal H-20 to C-1, -5, -9, and -10 revealed a *δ*-lactone ring present in the structure of **1**.

From the molecular formula of C_20_H_30_O_3_ of compound **1**, six double bond equivalents were calculated, which were assigned to the three rings (the lactone ring, ring B, ring C, and the three double bonds (the carbonyl group and the two carbon–carbon double bonds)). As a result, no additional ring existed in the structure of **1**, which determined that the original ring A in an isopimarane skeleton was opened in **1**. No HMBC correlation signal was observed from H-5 to the carbonyl carbon (C-3), revealing that the ring A opening occurred at the bond between C-3 and C-4, which was further confirmed by the presence of the *δ*-lactone ring present in the structure of **1**.

In order to further determine the absolute configuration, **1** was recrystallized in methanol to afford colorless crystals and was analyzed by X-ray crystallography. The crystallographic data of **1** (CCDC 1033449) are given in the Appendix A. These data can be obtained free of charge from The Cambridge Crystallographic Data Centre [22]. The structural refinement of the Cu *Kα* data of the crystal of **1** resulted in a Flack parameter of −0.1 (4) [23,24], allowing an explicit assignment of the absolute structure of **1** (Figure 3). The four chiral centers, C-5, -9, -10, and -13, were thus determined as *R*, *S*, *S*, and *S*, respectively. Accordingly, compound **1** was thus identified as 4*α*-hydroxy-3,4-*seco*-isopimara-7,15-diene-3,9*α*-olide, and it was given the trivial name of fladin B.

Compound **2**, [α${]}_{D}^{28}$ −2.14° (*c* 0.19, CHCl_3_), was obtained as colorless cubic crystals, and the molecular formula was determined to be C_20_H_28_O_2_ by positive HR-ESI-MS ([M + Na]^+^ *m/z* 323.1978 (calcd. 323.1987). The IR spectrum of **2** displayed a carbonyl group (*ν*_max_ 1718 cm^−1^). A comparison of the ^1^H and ^13^C NMR data suggested that the structure resembled compound **1**. When comparing the ^13^C NMR data, we observed that the signals at *δ*_C_ 74.2 and 34.1 in **1** disappeared. Instead, new signals appeared at *δ*_C_ 145.4 and 115.9, which were assigned to the double bond of Δ^4,19^ through the analysis of the 2D spectral correlation data of **2** (Figure 2). Accordingly, **2** was elucidated as 3,4-*seco*-isopimara-4 (18),7,15-triene-3,9*α*-olide, and it was given the trivial name of fladin C.

Compound **3**, [α${]}_{D}^{25}$ −12.9° (*c* 1.09, MeOH), was also obtained as colorless cubic crystals, and the molecular formula was determined to be C_20_H_30_O_2_ by positive HR-ESI-MS ([M + Na]^+^ *m/z* 325.2045, calcd. 325.2144). Compound **3** was found to be another closely related congener of **1**. In comparison with **1**, the signal at *δ*_C_ 74.2 disappeared, but a new signal was observed at *δ*_C_ 25.5 in the ^13^C NMR spectrum. Analysis of the HSQC and HMBC spectral data (Figure 2) of **3** assigned the signal of *δ*_C_ 25.5 belonging to the isopropyl group attached at C-5. Compound **3** was, thus, determined as 3,4-*seco*-isopimara-7,15-diene-3,9*α*-olide, and it was given a trivial name of fladin D.

### 2.2. X-ray Crystallographic Analysis

The molecular structures of the title compounds were built up from a bicycle [4.4.0] decene ring bearing *δ*-lactone, methyl, and vinyl substituents. The structural differences arose at C-5, with **1** having a 2-hydroxypropyl group, **2** having an isopropenyl group, and **3** having an isopropyl group. The X-ray analysis provided solid evidence to reveal the stereochemistry of this type of compound. In the molecule of **1**, the 2-hydroxypropyl group was *β*-oriented, as were the methyls of C-10 and C-13, whereas the *δ*-lactone ring and the C-13 vinyl group were α-oriented. In the skeleton, the *δ*-lactone ring (atoms C-1 to C-3/C-9/C-10/O-3) was confirmed to be a chair conformation, with C-3 and C-10 deviating from the best plane through atoms C-1/C-2/C-9/O-3 by -0.191 Å and 0.680 Å, respectively. The two trans-fused six-membered rings B (atoms C-5 to C-10) and C (atoms C-8 to C-9/C-11 to C-14) adopted half-chair and chair conformations, respectively. In the cyclohexene ring B, the deviations of atoms C-5 and C-6 from the best plane through atoms C-7/C-8/C-9/C-10 were 0.361 Å and 0.932 Å, respectively. In cyclohexane ring C, atoms C-9 and C-13 deviated by 0.585 Å and −0.674 Å from the best plane through atoms C-8/C-11/C-12/C-14.

The packing of compound **1** was characterized by a network of hydrogen bonds. A strong O—H···O hydrogen bond, namely, O1—H1···O2^i^ (Table 2), was formed via the hydroxyl group and the lactone oxo group running along the *b* axis direction (Figure 4).

### 2.3. Plausible Biogenetic Pathway of Fladins B–D *(**1**–**3**)*

Fladins B–D (**1**–**3**) possess a rare 3,4-*seco*-isopimarane skeleton with an unusual lactone moiety formed at C-9 and C-10. Hypothetically, they might be derived from isopimarane diterpenes, which are found abundantly in *Isodon* plants [11,12]. The original precursor of the diversified isopimarane-type diterpenes in *Isodon* plants could well be 3*β*-hydroxy-8,15-isopimaradiene (**4**), a rich isopimarane diterpenoid observed in several *Isodon* plants [10]. Given that background, a plausible biogenetic pathway of **1**–**3** originating from the natural precursor (**4**) is proposed (Figure 1) [10,25,26]. Compound **4** was initially converted to **5** through the oxidation of 3-OH and the epoxidation of the Δ^8,9^ double bond. By insertion of an oxygen atom between C-3 and C-4 of **5** (an enzymatic Baeyer–Villager reaction), compound **6** with a seven-membered lactone ring was formed. Ester hydrolysis of **6** by esterase produced diterpene carboxylic acid **7**, which underwent protonation in the epoxide group, followed by an SN2 reaction between the carboxylic acid group and the protonated epoxide group to provide **9**. Elimination of a hydroxy group in **9** yielded fladin B (**1**), which eliminated another hydroxy group to generate fladin C (**2**). Compound **2** underwent a hydrogenation reduction on the Δ^4,19^ double bond to produce fladin D (**3**).

### 2.4. Bioactivity Evaluation of Fladins B–D *(**1**–**3**)*

Previously, two diterpenoids isolated from *I. flavidus* were reported to have inhibition activity against *Trichophyton rubrum* with MIC values around 25.5–197 μM [10]. In order to find out whether our isolates were the active ingredients of this medicinal plant, we evaluated the antifungal potential against two athlete’s foot fungal strains (*T. rubrum* and *T. mentagrophytes*) of compounds **1**–**3**. However, no inhibitory activities against the two fungi were observed at concentrations below 20 µg/mL (62.9–66.7 µM). Cell viability was also measured for each pure compound to determine the cytotoxic activities of the compounds, but no compound was found to show inhibitory effects against HCT116 and A549 cancer cells at concentrations below 20 µg/mL (62.9–66.7 µM) as well. The compounds were then tested for their antiviral potential using our established pseudovirus screening assay system. No compounds displayed antiviral activities against HIV entry below the concentration of 20 µg/mL (62.9–66.7 µM), and compounds **1** and **3** also showed no antiviral activities against Ebola entry below the concentration of 20 µg/mL (62.9–66.7 µM). Interestingly, only fladin C (**2**) showed 81% inhibition against the Ebola virus at a concentration of 20 µg/mL (66.7 μM). To the best of our knowledge, there have been no reports of *seco*-isopimarane-type diterpenes having antiviral activity.

## 3. Discussion

Plants from the genus *Isodon* are claimed to be a rich source of bioactive diterpenes with a diversity of carbon skeletons. The folk medicinal uses of *I. flavidus* [22] and our previous discovery of antifungal diterpenes from *I. flavidus* [10] gave rise to our great interest in this medicinal plant. Our ongoing research of the methanolic extract of the twigs and leaves of *I. flavidus* led to the identification of three new 3,4-*seco*-isopimarane diterpenes [fladins B–D (**1**–**3**)]. Their structures were determined based on the analysis of the comprehensive spectroscopic data, and the absolute configuration of fladin B (**1**) was determined by X-ray crystallographic data. The structures of **1**–**3** with a new *δ*-lactone ring system between C-3 and C-9 were formed through the rearrangement at ring A of an isopimarane skeleton by the bond break of C-3 and C-4. In the crystal of **1**, molecules were linked to each other by intermolecular O-H···O bonds, forming chains along the *b* axis. The differences among the structures of **1**–**3** were only in the isopropyl group presented with a hydroxy group at C-4 for fladin B (**1**), a Δ^4,19^ double bond for fladin C (**2**), and no substituent at C-4 for fladin D (**3**). Accordingly, compounds **1**–**3** were identified as 3,4-*seco*-isopimarane-type diterpenes. The 3,4-*seco*-isopimaranes could be produced from the naturally occurring precursor 3*β*-hydroxy-8,15-isopimaradiene (**4**) in the *I. flavidus* plant through the biogenetic pathway outlined in Figure 1.

Natural isopimaranes were demonstrated with broad biological activities, including antimicrobial activities. For example, our previous isolated 3,4-*seco*-isopimarane diterpene fladin A, which has a cyclic ether group formed between C-4 and C-9, showed inhibition activity against *T. rubrum* with a MIC value of 62.5 μg/mL [10]. In the present study, the cytotoxic, antifungal, and antiviral activities of **1**–**3** were investigated. Although the three compounds possessed similar structures, only compound **2** was observed to inhibit Ebola entry at the concentration of 20 µg/mL (66.7 µM). The results revealed that the antiviral potential of 3,4-*seco*-isopimarane-type diterpenes could be improved by further structural modification on some specific functional groups.

## 4. Materials and Methods

### 4.1. General Experimental Procedures

One-dimensional and two-dimensional NMR spectra of fladins B and D were recorded on a JEOL JNM-ECS400 (400 MHz) spectrometer (JEOL, Tokyo, Japan), and those of fladin C were recorded on a JEOL ECX 500M (500 MHz) spectrometer (JEOL, Tokyo, Japan). Chemical shifts (*δ*) were expressed in ppm, and coupling constants (*J*) were reported in Hz. All NMR experiments were obtained by using standard pulse sequences supplied by the vendor. Optical rotation was measured with a Perkin-Elmer model 241 polarimeter (Perkin Elmer, Waltham, MA, USA). IR spectra were recorded on a Bruker VECTOR22 spectrophotometer (Bruker, Rheinstetten, Germany) with KBr pellets. High-resolution electrospray ionization mass spectroscopy (HR-ESI-MS) was recorded on a VG Autospec-3000 spectrometer (VG, Manchester, England). Column chromatography was performed with silica gel (200–300 mesh; Qingdao Marine Chemical, Inc., Qingdao, China). Thin-layer chromatography (TLC) was performed on glass plates coated with silica gel GF_254_ (Qingdao Marine Chemical Inc.). All solvents, including petroleum ether (60–90 °C), were distilled prior to use. X-ray crystallographic data were obtained on a Bruker *APEX*-II CCD instrument (Bruker, Rheinstetten, Germany) using Cu K*α* radiation.

### 4.2. Plant Materials

The collection of the twigs and leaves samples of *I. flavidus* was made in September 2012 in Leishan county, Guizhou, China. This plant species was authenticated by Professor De-Yuan Chen of the Guizhou University of Traditional Chinese Medicine and deposited in the same university with the accession number No. 20120903.

### 4.3. Extraction and Isolation

The air-dried milled plant material of *I. flavdus* (5.5 kg) was extracted with 95% methanol (3 × 10 L) at room temperature and concentrated in vacuo to give 556 g of extract. The methanol extract was processed as previously described [10] to provide six fractions (A–F). Fraction C was chromatographed over an additional silica gel column, which was developed by gradient elution (CHCl_3_/MeOH gradient, from 5:1 to 1:1) to provide fladin B (**1**, 5 g) and sub-fractions (C_1_–C_4_). Fladin B (**1**) was then recrystallized from methanol at room temperature to provide colorless crystals suitable for X-ray crystallographic analysis. Sub-fraction C_3_ was separated over a Sephadex LH-20 column (CHCl_3_/MeOH gradient, 1:1) to afford fladins C (**2**, 32 mg) and D (**3**, 5 mg).

#### 4.3.1. Fladin B (1)

Colorless cubic crystals (MeOH); m.p. 155–157 °C; [α${]}_{D}^{28}$ −8.59° (*c* 0.33, CHCl_3_); UV (CHCl_3_) *λ*_max_ (log *ε*) 244 (0.94) nm; IR (KBr) *ν*_max_ 3450, 2970, 2928, 1693, 1462, 1424, 1383, 1367, 1153, 842 cm^−1^; ^1^H and ^13^C NMR, see Table 1; HR-ESI-MS ([M + H]^+^ *m*/*z* 319.2269, calcd. 319.2273 for C_20_H_31_O_3_).

#### 4.3.2. Fladin C (2)

Colorless cubic crystals (MeOH); m.p. 146–148 °C; [α${]}_{D}^{28}$ −2.14° (c 0.19, CHCl_3_); UV (CHCl_3_) λ_max_ (log ε) 243 (0.93) nm; IR (KBr) ν_max_ 3454, 2960, 2922, 1718, 1638, 1434, 1386, 1328, 1033, 853 cm^−1^; ^1^H and ^13^C NMR, see Table 1; HR-ESI-MS ([M + Na]^+^ *m*/*z* 323.1978, calcd. 323.1987 for C_20_H_28_O_2_Na).

#### 4.3.3. Fladin D (3)

Colorless cubic crystals (MeOH); m.p. 149–151 °C; [α${]}_{D}^{25}$ −12.9° (*c* 1.09, MeOH); UV (CHCl_3_) *λ*_max_ (log *ε*) 223 (0.93) nm; IR (KBr) *ν*_max_ 3432, 2955, 1728, 1719, 1712, 1470, 1346, 1065, 539 cm^−1^; ^1^H and ^13^C NMR, see Table 1; HR-ESI-MS ([M + Na]^+^ *m*/*z* 325.2045, calcd. 325.2144 for C_20_H_30_O_2_Na).

### 4.4. Single Crystal X-ray Data of Fladin B (1)

Crystal data of **1** (from MeOH): space group *P*21, C_20_H_30_O_3_, M = 318.44, *a* = 11.4575 (12) Å, *b* = 6.3205 (6) Å, *c* = 12.5407 (13) Å, *α* = 90.00°, *β* = 108.004 (4)°, *γ* = 90.00°, *V* = 863.69 (15) Å^3^, *T* = 100 (2) K, Z = 2. μ (CuKα) = 0.632 mm^−1^, 4892 reflections measured, 2556 independent reflections (*R_int_* = 0.0574). The final *R*_1_ values were 0.0936 *(I* > 2*σ* (*I*)). The final *w*R (*F^2^*) values were 0.2355 (*I* > 2*σ*(*I*)). The final *R*_1_ values were 0.0948 (all data). The final *w*R (*F^2^*) values were 0.2382 (all data). The goodness of fit on *F*^2^ was 1.108. The Flack parameter was −0.1 (4). The Hooft parameter was 0.10 (15) for 917 Bijvoet pairs. Data were deposited in the Cambridge Crystallographic Data Centre with No. CCDC 1033449.

### 4.5. Bioactivity Evaluation

Cytotoxicity assays were performed against human colon (HCT116) and lung (A549) cancer cell lines by using Sulforhodamine B (SRB, Fluka, Cat. No. 86183, Buchs, Switzerland), as previously reported [27,28].

The antifungal activity was evaluated against *Trichophyton rubrum* (ATCC MYA-4438) and *T. mentagrophytes* (ATCC 28185) (obtained from the Institute of Dermatology, Chinese Academy of Medical Science, Nanjing, China). The antifungal assays were performed using the broth microdilution method, as described in M38-A2, with modifications [29,30,31].

Compounds **1**–**3** were then evaluated for their anti-HIV and anti-Ebola activities using our previously established “One-Stone-Two-Birds” assay evaluation system as previously described with a modified procedure [32,33].

## 5. Conclusions

In conclusion, we isolated three novel compounds [fladins B-D (**1**-**3**)] from *I. flavidus* belonging to 3,4-*seco*-isopimarane diterpenes characterized by a unique *δ*-lactone ring formed between C-3 and C-9. The biological evaluation revealed that only **2** could inhibit viral replication against Ebola entry. The differences among the structures of **1**–**3** occurred only in the C-5 isopropyl group. The presence of the carbon–carbon double bond at the isopropyl group could be a key contributor to the antiviral activity of a 3,4-*seco*-isopimarane. Further structural modification is thus needed to reveal the structure–activity relationship of 3,4-*seco*-isopimaranes in order to improve their biological activities.

## Data Availability

The data associated with this study are included in this published article. Appendix A with original NMR, IR, and MS spectra for compounds **1**–**3** are included in the article. Additional files are available from the corresponding authors upon reasonable request.

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
