# Peer review of "3,4-Seco-Isopimarane Diterpenes from the Twigs and Leaves of Isodon Flavidus"

_molecules, 2022, doi:10.3390/molecules27103098_

Round 1

Reviewer 1 Report

The work described in this manuscript is very interesting and relevant because it shows the structure of new bioactive compounds in tissues of Isodon flavidus. The work is clearly justified and well performed. Sophisticated and adequate equipment was used. The data are well described and discussed. The presentation of the manuscript, including the English presentation, is also very good. I have only minor comments about this manuscript.

-The title of the manuscript should contain the term “3,4-seco-isopimarane diterpenes” because the novelty of the study relays on this type of compounds.

-Perhaps, the word “targets” is unnecessary in line 25.

-Please, review the use of the term “lead compounds” in line 32. It is confusing.

Author Response

Dear Reviewer,

We thank you very much for the comments and suggestions. Please kindly see the attachment of our response. All of the changes are used with the “track changes” function.

Reviewer 2 Report

Comments:

The work “Isopimarane diterpenoids from the twigs and leaves of Isodon 2 flavidus” is promising as progression of searching of natural compound throughout the world and the author did the convincing work, but needs to be clarified as follows:

  1. The hypothesis should be clearly elaborated in the introduction part with exact references.
  2. Result should be clearly described. Authors said about 1D and 2D NMR spectra but in the whole manuscript I couldn’t find the pick. Author should provide the pick of graph generated. Author claims, there have been no reports of seco-isopimarane-type diterpenes having antiviral activity. In this concern I have confused, needs to be clarify it???
  3. Discussion section is poorly written and needs to be improved.
  4. Conclusion should be based on the results and needs to be rewritten.
  5. There are also numerous typographical faults and grammatical errors to contend with.  This manuscript should check by a native English-speaking colleagues or need professional English editing service.

 Therefore, I recommended it for major revision.

Author Response

(The authors gave the same response as above.)

Reviewer 3 Report

I found the manuscript " Isopimarane diterpenoids from the twigs and leaves of Isodon 2

flavidus" is interesting, isolation, structural identification as well as biological evaluation of

these novel 3,4-seco-isopimarane diterpenes from the plantThe topic is of current interest and suited for the journal; anyways, some minor modifications of the submitted paper are recommended before publication.

ABSTRACT

Over all abstracrt is well writteb but Author should add more details about biological activities.

Introduction

Authors must be describe about targeted biological activities, here just mentioned like antiviral or antifungal.

2.4. Bioactivity evaluation of fladins B-D (1-3)

Here is also need to add some specific species name.

Over all manuscript written very well

I am recomeding this manuscript for publication after revision.

Author Response

(The authors gave the same response as above.)

Reviewer 4 Report

Please see attached pdf file.

Author Response

(The authors gave the same response as above.)

Round 2

Reviewer 2 Report

Authors have answered all the concern raised by me and the present form of manuscript looks promising and I am recommending  for acceptance for publication.